# Human Brain Models of Intellectual Disability: Experimental Advances and Novelties

**DOI:** 10.3390/ijms23126476

**Published:** 2022-06-09

**Authors:** Nona Laura Lisa Merckx, Hilde Van Esch

**Affiliations:** 1Laboratory of Stem Cell and Developmental Neurobiology, VIB-KU Leuven Center for Brain & Disease Research, 3000 Leuven, Belgium; 2Center for Human Genetics, University Hospitals Leuven, 3000 Leuven, Belgium; hilde.vanesch@kuleuven.be; 3Laboratory for the Genetics of Cognition, Department of Human Genetics, KU Leuven—University of Leuven, 3000 Leuven, Belgium

**Keywords:** intellectual disability (ID), pluripotent stem cells, 2D models, organoids, CRISPR/Cas9

## Abstract

Intellectual disability (ID) is characterized by deficits in conceptual, social and practical domains. ID can be caused by both genetic defects and environmental factors and is extremely heterogeneous, which complicates the diagnosis as well as the deciphering of the underlying pathways. Multiple scientific breakthroughs during the past decades have enabled the development of novel ID models. The advent of induced pluripotent stem cells (iPSCs) enables the study of patient-derived human neurons in 2D or in 3D organoids during development. Gene-editing tools, such as CRISPR/Cas9, provide isogenic controls and opportunities to design personalized gene therapies. In practice this has contributed significantly to the understanding of ID and opened doors to identify novel therapeutic targets. Despite these advances, a number of areas of improvement remain for which novel technologies might entail a solution in the near future. The purpose of this review is to provide an overview of the existing literature on scientific breakthroughs that have been advancing the way ID can be studied in the human brain. The here described human brain models for ID have the potential to accelerate the identification of underlying pathophysiological mechanisms and the development of therapies.

## 1. Intellectual Disability

Intellectual disability (ID), previously termed “mental retardation”, is defined as a disorder with onset during the developmental period that includes both intellectual and adaptive functioning deficits in conceptual, social, and practical domains. The overall general population prevalence of ID is approximately 1% and varies by age. Diagnosis of ID requires the following three criteria:1Deficits in intellectual functions, such as reasoning, problem solving, planning, abstract thinking, judgment, academic learning, and learning from experience, confirmed by both clinical assessment and individualized, standardized intelligence testing.2Deficits in adaptive functioning that result in failure to meet developmental and sociocultural standards for personal independence and social responsibility. Without ongoing support, the adaptive deficits limit functioning in one or more activities of daily life, such as communication, social participation and independent living, across multiple environments, such as home, school, work and community.3The onset of intellectual and adaptive deficits during the developmental period. ID applies a heavy burden on the affected individuals, families, the society and the health care system. The extremely heterogeneous nature of ID is a consequence of the wide-ranging underlying etiology. Severe ID can be identified by delayed motor, language and social milestones during the first 2 years of life, while mild ID may not be identifiable until school age when academic learning starts [1].

When a detailed history and physical examination gives reason to suspect a specific clinical diagnosis, confirmation by molecular laboratory analysis is required. ID is influenced both by genetic as well as environmental factors that affect the development and functioning of the nervous system, prenatally, perinatally, or postnatally or a combination thereof. Conventional cytogenetics, such as karyotyping and fluorescence in situ hybridization (FISH), allow the identification of about 15% of ID causes [2]. The advent of chromosomal microarrays (CMA) and whole exome sequencing (WES), two clinical tests frequently applied in the diagnosis of ID, has increased the diagnostic yield of neurodevelopmental disorders (NDDs) to 15–20% and 30–43%, respectively [3]. Genetic causes are more frequently observed in the group of severe ID and include chromosomal aneuploidies, submicroscopic chromosomal rearrangements, copy number variations and monogenic disorders. Clinically, ID can be divided into nonsyndromic and syndromic forms, depending on the absence or presence of other clinical features [4].

## 2. Novel Models of ID

ID is ideally studied in the human brain, however, studying human brain samples imposes major challenges. Fresh human brain samples are hard to obtain and the majority of the human samples are obtained from the end stages of the disease or post-mortem, prohibiting the correct study of the development of ID pathology in patients [5]. In addition, the lack of adequate central nervous system (CNS) models might explain the low success rate of specific CNS drug development [6]. Over the past decades, novel and modernized experimental tools to study NDDs, among which ID, broadened our knowledge about underlying molecular mechanisms and potential therapies. Here, we aim to review the current models and techniques that have been revolutionizing ID studies.

### 2.1. Pluripotent Stem Cells

Embryonic development is characterized by a small population of pluripotent stem cells (PSCs) that give rise to all cells in the mature organism. Human PSCs cultured in vitro, also referred to as human embryonic stem cells (hESCs), are broadly utilized to generate any somatic cell type [7,8,9,10,11]. The capacity of PSCs to differentiate into a large range of specified cell types has enabled access to previously hard to obtain samples, such as human cortical neurons [12,13,14,15]. However, the fact that hESCs originate from early-stage embryos created ethical concerns for their application in clinical trials [16]. Pioneering studies by S. Yamanaka and co-workers enabled the reprogramming of somatic cells into a PSC-like state. For this purpose, fibroblasts from skin biopsies and peripheral blood derived T lymphocytes are transformed into human-induced PSCs (hiPSCs) by ectopically expressing Yamanaka factors OCT3/4, SOX2, KLF and c-MYC [17,18]. The scientific breakthrough of hiPSCs provides a renewable supply of patient-derived cells to study the pathophysiology of diseases and to design personalized therapies. The hiPSCs are a useful tool to study individual patients with known or unknown genetic mutations considering the particular heterogeneous nature of ID [19]. In this way, iPSC lines and models have been established for a large number of ID disorders, including Fragile X syndrome, Rett syndrome, Dravet syndrome, Phelan–McDermid syndrome, Miller–Dieker syndrome, Angelman syndrome Alexander disease, Timothy syndrome, Williams–Beuren syndrome, Prader–Willi syndrome, microcephaly and others [20].

### 2.2. Human Brain Models In Vitro

Models that rely on PSC technology to study the human brain are constantly evolving. Several differentiation protocols have shown to successfully convert the PSCs into functional neurons in 2D as monolayers or 3D as organoids [12,13,15,21,22]. The general concept behind these protocols is exposing PSCs to growth factors and morphogens to mimic neural induction that takes place during embryonic development. Adding patterning factors that target Smad, Wnt and Shh signaling drives the differentiation towards specific brain regions. Brain organoids are 3D cultures that contain multiple types of cells and cytoarchitectures and resemble human fetal brain structurally and functionally. These organoids are being used increasingly to model brain development and disorders, including ID syndromes. Protocols to generate organoids of cerebellar, hypothalamic/pituitary, hippocampal, thalamic, brainstem, midbrain, spinal cord and choroid plexus identity have been described to date [23]. More recent, region-specific organoids have been developed that can further be used to create more complex assembloids. These assembloids can help to recapitulate inter-regional and inter-cellular interactions as well as neural circuitry development by combining multiple brain regions and/or cell lineages [24]. In an alternative approach, neural induction is achieved without the use of exogenous growth factors or morphogens. By supplying improved growth conditions and the environment necessary for intrinsic cues to influence development, cells composed of multiple regional identities of the brain are generated [22,25]. In vitro human brain models chronologically recapitulate key developmental milestones of fetal and early neonatal brains [26]. Therefore, many characteristics of brain development can be studied, such as cell cycle/proliferation, mitosis, neurogenesis, gliogenesis, apoptosis, migration, morphology, maturation and network activity (Figure 1). Furthermore, recording neuronal function in ID models is essential for understanding pathophysiology and drug development. The electrophysiological methods that provide a read out of neuronal function underwent rapid advancements in the past decades. The classical functional method, patch clamping, provides electrophysiological details of individual neurons with high temporal resolution, but lacks the larger scale resolution of network connectivity and dynamics. Alternatively, the calcium status of a cell can be measured, which is associated with the generation of action potentials. The calcium status can be imaged in groups of neurons simultaneously using a calcium indicator such as GCaMP, permitting a larger-scale recording of neuronal and synaptic activity in neural circuits. As a tradeoff, high temporal resolution is lost and the technique depends on imaging capacities. The emergence of microelectrode arrays (MEAs) overcomes the previous limitations by combining the high temporal resolution of patch clamping with the large-scale network resolution of calcium imaging. MEAs measure the extracellular potentials of a relatively large number of neurons, providing the parameters of network connectivity. It is a challenge to analyze 3D models by traditional calcium imaging and MEAs as they are limited to recording almost exclusively the outer edges of organoids. In addition, single cell resolution is lost in MEAs, which makes it impossible to correlate the output to individual neuron subtypes. Increased electrode density and 3D-MEAs are currently under development, foreseeing future technologies that provide single cell activity analysis on a larger scale. This could enable the study of individual neuronal subtype connectivity within different subregions of in vitro models, such as specific neocortex layer neurons that are interconnecting different subregions of organoids [27,28,29,30]. In vitro models can be utilized to study the effect of genetic defects or environmental exposure, such as substances, viral infections (ZIKA virus) or hypoxia on brain development and function [31]. These models provide endless approaches to study ID in great detail. Nonetheless, in vitro human brain models convey serious limitations to correctly model ID. Neuronal circuits remain relatively immature, encompassing only early stages of fetal brain development. Human cortical neurons take months to establish proper synaptic connectivity, which seriously challenges culture maintenance. Essential features of cognitive development, such as neuronal maturation, synapse formation and network integration, are incomplete or absent [32]. The lack of the physiological environment of the in vivo brain results in activated cellular stress, displayed by increased glycolysis and ER stress and inhibited cell subtype specification [33]. A novel model to overcome these issues is discussed in the next section.

### 2.3. Human Brain Models In Vivo—Xenotransplantation

A cutting edge model to overcome the experimental issues of in vitro human brain models is the introduction of an in vivo model using xenotransplantation (Figure 2). Grafting the in vitro hiPSC-derived neurons in the mouse neonatal cortex enables their further development in vivo, in a more physiological condition. Briefly, hiPSC-derived cortical neurons are differentiated in vitro followed by dissociation into single cells and injection into the lateral ventricles of neonatal mice. After transplantation, human neurons integrate into the host neuronal circuits of the cerebral cortex, grow axons and dendrites and functionally participate in synaptic transmission. This mouse–human chimeric approach recapitulates key milestones of human neuronal development. Grafted neurons display coordinated morphological and physiological maturation, robust dendritic spine dynamics and functional synaptic plasticity. This model enables a longer follow-up period of human neuronal maturation up to at least 11 months, more advanced than reported so far for in vitro models. This is especially important considering the remarkable prolonged maturation period of human neurons, which has a strong cell-intrinsic component [32,34]. Moreover, the model encompasses more appropriate physiological conditions close to the human brain environment, which rescues the in vitro activated cellular stress and increases cell subtype specificity [33]. This promising experimental approach provides a unique opportunity to model ID in vivo. Although significantly enhanced, one notable limitation remains the incomplete maturation of neurons. Not surprisingly, grafted neurons still do not reach the same morphological level of maturation that is observed in neurons of the adult human brain, such as their larger cell bodies and more arborized dendrites [32]. Therefore, the in vivo study of ID, additionally, relies on other in vivo models to completely understand the pathophysiology.

### 2.4. Gene Editing Tools

When ID is caused by genetic defects, studies used to compare patient-derived cell lines to healthy controls. The presence of inter-individual differences in the genetic background used to complicate the comparison of patient samples to controls. Today, isogenic control lines are obtained by correcting the genetic defects using gene editing tools, such as CRISPR/Cas9, transcription activator-like effector nucleases (TALENs) and zinc-finger nucleases (ZFNs) [35,36]. The Cas9 nuclease can be used for editing the genomic sequence by inducing targeted DNA double-strand breaks (DSBs), which are corrected by non-homologous end-joining (NHEJ) and to a lower extent homologous recombination (HR). In addition to multiple applications in fundamental research, the enormous potential of the CRISPR/Cas9 system is now exploited in the development of gene therapies for several genetic neurodevelopmental disorders [37,38]. In 2018, the first gene therapy for an inherited genetic disease was approved by the EMA for marketing. Luxturna, a gene therapy for the previously incurable *RPE65*-Leber congenital amaurosis (LCA), which involves blindness, improves the ability to detect light. By a subretinal injection of an adeno-associated viral (AAV) vector with a functional copy of *RPE65*, vision improved with a favorable benefit-to-risk profile [39]. This development underscores the importance and potential of designing gene therapies that aim to correct causal genetic mutations, which is often the case in ID. Using AAVs to replace a non- or dysfunctional gene creates the opportunity to treat ID. An obstacle to applying gene therapy for ID might be the difficulty of vector delivery to the brain. Serotype AAV9 can cross the blood-brain barrier with systemic administration. The efficacy of CNS transduction can be increased by direct delivery into the cerebrospinal fluid (CSF), reducing the exposure of peripheral tissues. The current challenges of AAV gene therapy include the unwanted host immune response and a maximum gene sequence size of 4.4 kb. An important question regarding ID is whether there is a defined therapeutic window in which we must intervene to see clinical benefit [40]. ID caused by dysfunctional genes that are essential during early stages of brain development might not be appropriate targets for gene therapy since therapeutic intervention will only be efficient when initiated early. Additionally, genome editing techniques comprise the risk of off-targeted genome or epigenome events [41].

## 3. Modelling ID in Practice

The described models and technologies are applied to study the pathophysiology of a number of diseases characterized by ID. We here give a few examples of how these technologies have been put in practice to study ID and design therapies.

### 3.1. Fragile X Syndrome

Fragile X Syndrome (FXS) is the most prevalent form of inherited ID characterized by a complex neurodevelopmental phenotype involving moderate to severe cognitive impairment, epilepsy, auditory hypersensitivity, repetitive behavior and social withdrawal. FXS is X-linked and caused by a CGG triplet repeat expansion in the 5′ UTR region of the fragile X messenger ribonucleoprotein 1 (*FMR1*) gene. If the expansion is larger than 200 repeats, the promoter of *FMR1* is hypermethylated resulting in gene inactivation and absence of the encoding protein FMRP [42]. Hypermethylation starts approximately at the end of the first trimester, after which the *FMR1* promoter becomes gradually hypermethylated leading to increasing *FMR1* silencing. Therefore, FMRP is still present during early developmental processes but absent during late developmental stages, such as neurogenesis from progenitors, migration and synaptogenesis. The gradual gene silencing is unique to humans and not recapitulated in *Fmr1* knock out (KO) mice, indicating the requirement for a human model to correctly study FXS pathophysiology. Several studies have suggested absence of FMRP leads to reduced GABA-mediated inhibition, which causes neuronal hyperexcitation, behavioral hyperactivity and epilepsy, while other studies argue that abnormal ion channel activity and firing pattern, decreased neurotransmitter release and reduced synaptogenesis are the cause. These theories remain to be confirmed or refuted, for which hiPSC models may provide the solution [43,44]. In the past, an FXS clinical trial using mGluR5 agonists had to be discontinued after failure of improving the disease symptoms [45]. In hindsight, we now know from the FXS patient derived hiPSC model that the altered mGluR-dependent signaling in the FXS rodent model is not recapitulated [43]. Furthermore, the FXS patient-derived hiPSC model taught us that in vitro differentiated neural progenitors have affected calcium signaling via AMPARs, contributing to the aberrances in neural circuit formation and function [46]. Moreover, recently 2D and 3D in vitro human FXS models and CRISPR/Cas9 induced wildtype counterparts were generated, which revealed altered neuronal and glial gene expression, increased network activity and increased excitation/inhibition ratio with increased size and number of cortical plates. The results suggest that FMRP is responsible for the neuronal glial balance in the cerebral cortex, possibly regulated by the GFK-3β/Notch pathway. The arrival and further development of FXS 2D and 3D brain models can provide the opportunity to explore effective treatments in humans [47].

### 3.2. Rett Syndrome

Rett syndrome (RTT) is caused by haploinsufficiency of MECP2 as a result of different mutations. Disease onset is characterized by developmental stagnation and regression starting at 6 to 18 months of age after a period of normal prenatal and postnatal development. The syndrome occurs almost exclusively in females, as the mutations are lethal in males in general. RTT symptoms vary among patients who can display ID, a regression of acquired skills, loss of speech, stereotypic movements, microcephaly, seizures and motor problems [48]. Phenotypic variability is attributed to the type and location of the mutation in *MECP2*. There are over 500 pathogenic mutations in *MECP2* reported, and several of these have now been studied using patient-derived iPSC models reprogrammed from fibroblasts and comparing them to healthy controls. More recently also isogenic controls have been obtained by CRISPR/Cas9 correction of the patient mutation in patient derived cell lines. The 2D differentiation models of RTT hiPSCs-derived cortical neurons revealed smaller soma, diminished dendritic trees, a reduced number of dendritic spines, decreased neuronal activity and synaptic dysregulation [49,50,51,52,53,54]. Screening pharmacological compounds in RTT 3D organoids compared to isogenic controls, identified compounds that reversed the neuropathologic phenotypes in vitro [52]. Transcriptomic studies in hiPSC derived RTT neurons displayed a global transcriptional repression [55]. *MECP2* is a suitable gene target of gene therapy, since phenotypic reversal in RTT mice was achieved by delayed gene restoration in both immature and mature adult animals, providing a relatively long therapeutic window [56]. Importantly, the developmental regression present in RTT occurs only 6 to 18 months after birth. Therefore, studying postnatal developmental timepoints might be crucial to understand the mechanism that cause the late onset of symptoms. The in vivo xenotransplantation model of human neurons provides the opportunity to investigate the later timepoints that (so far) have been impossible to model in vitro.

### 3.3. MECP2 Duplication Syndrome

MECP2 duplication syndrome (MDS) is caused by an extra copy of the MECP2-locus involving the entire gene, resulting in severe to profound ID, infantile hypotonia, mild dysmorphic features, poor speech development, autistic features, seizures, progressive spasticity and recurrent infections [57,58]. In general, only males are affected by MDS, while female carriers mostly show complete skewing of X-inactivation towards the normal allelic locus. MDS demonstrates a morphological and functional cellular phenotype that is in sharp contrast with RTT neurons: MDS hiPSC-derived cortical neurons in 2D showed increased dendritic arborization and complexity associated with increased number of glutamatergic synapses, altered spine maturation, significantly increased frequency of activity and synchronized bursts [59]. The underlying mechanisms of MDS pathophysiology remain largely unknown.

### 3.4. Williams–Beuren Syndrome

The Williams–Beuren syndrome (WBS) is caused by contiguous gene deletion of the Williams–Beuren syndrome critical region (WBSCR) that encompasses approximately 25 genes at chromosome 7q11.23, including the elastin gene (*ELN*). WBS occurs in both males and females and is characterized by mild to moderate ID, a specific cognitive profile (strengths in verbal short-term memory and language and extreme weakness in visuospatial construction), unique personality (overfriendliness, empathy, generalized anxiety, specific phobias and attention deficit disorder), growth abnormalities, cardiovascular disease, distinctive facies, connective tissue abnormalities and endocrine abnormalities [60]. Studies of the underlying pathophysiology of WBS largely relied on animal models in vivo, which gave insight in the cellular and molecular phenotypes in a non-human context [61]. WBS human tissue samples mainly explored the effect on cell types unrelated to brain development and function [62,63]. Several studies compared in vitro WBS patient-specific hiPSCs to non-isogenic wildtype hiPSCs, exploring the affected human-specific gene networks in differentiated cortical neurons. WBS-derived neurons have significantly prolonged action potential repolarization times suggesting a defect in potassium channel conductance. In addition, WBS neurons differentially expressed genes implicated in neurotransmitter receptors, synapse assembly and potassium channel complexes. Layer V/VI cortical neurons derived from in vitro WBS iPSCs are characterized by longer total dendrites, increased numbers of spines and synapses, aberrant calcium oscillation and altered network connectivity [64,65,66].

### 3.5. Prader–Willi Syndrome

The Prader–Willi syndrome (PWS) is a contiguous gene disorder caused by the loss of function of the paternally contributed chromosomal region 15q11-q13. The 15q11–13 chromosomal region is regulated by genomic imprinting, an epigenetic phenomenon in which the expression of an allele is determined by the parent of origin. The locus contains 6 small nucleolar RNA genes and 6 protein-coding genes (*MKRN3*, *NDN*, *NPAP1*, *SNURF-SNRPN* and melanoma antigen gene family member L2 *MAGEL2*). PWS is characterized by severe hypotonia and feeding difficulties in early infancy, followed in later infancy or early childhood by excessive eating and gradual development of morbid obesity (unless eating is externally controlled). Motor milestones and language development are delayed. All individuals have some degree of cognitive impairment, ID becomes generally evident when the individual reaches preschool age. A distinctive behavioral phenotype (with temper tantrums, stubbornness, manipulative behavior and obsessive-compulsive characteristics) is common. Hypogonadism is present in both males and females and manifests as genital hypoplasia, incomplete pubertal development and, in most, infertility [67]. PWS hiPSC derived neuronal culture models have impaired secretory granule (SG) abundance and neuropeptide production, which corresponds to the *Magel2* loss of function mouse model. In addition, PWS patient derived dental pulp stem cells (DPSCs) differentiated into neurons revealed severely defected retromer and WASH-dependent endosomal recycling of cargo that includes the components of SG [68]. Importantly, the imprinted status of genes can change during the induced reprogramming of somatic cells into iPSCs. A study that reprogrammed fibroblasts into iPSCs detected a maternal to paternal epigenotype switch at the PWS imprinting center (IC) in the control iPSC line. Additionally, the ability to influence the epigenetic status of *MKRN3*, one gene in the 15q11-q13 region, was lost. This should be taken into account when using iPSCs to model imprinting diseases [69].

### 3.6. Angelman Syndrome

The Angelman syndrome (AS), the counterpart of PWS, is a contiguous gene disorder also caused by the loss of function of the maternal 15q11-q13 chromosomal region. AS results from the loss of function of the *UBE3A* gene, a gene housed within this region. Due to genomic imprinting, *UBE3A* is solely expressed from the maternal allele in neurons. The paternal allele is silenced in these cells by the reciprocal expression of a long, non-coding antisense RNA3. Therefore, the loss of the maternal allele of *UBE3A* results in the loss of *UBE3A* mRNA and protein in neurons [70]. AS is characterized by severe developmental delay or intellectual disability, severe speech impairment, gait ataxia and/or tremulousness of the limbs, and unique behavior with an apparent happy demeanor that includes frequent laughing, smiling and excitability. Microcephaly and seizures are also common. Developmental delays are first noted at around the age of six months; however, the unique clinical features of AS do not become manifest until after the age of one year [71]. AS patient derived hiPSCs differentiated into cortical neurons have been compared to non-isogenic healthy controls as wildtype, and in which *UBE3A* was knocked out using CRISPR-Cas9 or knocked down using antisense oligonucleotides (ASOs). During early timepoints, AS neurons behave as controls, but after 6–8 weeks in vitro deficits become apparent. AS neurons have a more depolarized resting membrane potential (RMP), immature action potential (AP) firing, decreased spontaneous excitatory synaptic activity and reduced capacity for activity-dependent synaptic plasticity. Interestingly, these phenotypes are rescued by unsilencing paternal *UBE3A* expression [70].

## 4. Challenges and Future Perspectives

Despite the fact that these novel ID models and techniques have proven to be promising tools to unfold ID pathophysiology, challenges remain that require improvement:Individuals suffering from ID often carry unique or rare mutations. Reprogramming and differentiating patient derived cell lines involve high costs, time and expertise [20].Given that ID comprises cognitive testing and other psychiatric comorbidities, behavioral assays still rely on animal models. However, animal models lack specific cellular features, such as the complexity and slow maturation properties, that are unique to the human brain [72].Late timepoints of ID etiology are not obtainable due to the incomplete maturation of hiPSC-derived neurons. Especially for diseases with late onset of symptoms, such as RTT, this poses a problem, as the underlying mechanisms preceding late onset cannot be studied up until now.Although iPSCs can be used to model imprinted diseases, one question is whether the genome-wide imprinting status is conserved during the epigenetic rewiring that takes place during somatic reprogramming. iPSCs are known to have defective imprinting, even using different reprogramming procedures. This should be taken into account when using iPSCs to model diseases caused by imprinting defects [69,73].In vitro models experience metabolic and endoplasmic reticulum (ER) stress on the cultured cells, which might hamper correct analyses and interpretation of findings. Moreover, in vitro models do not reproduce the complexity of the in vivo brain. Xenotransplantation of in vitro cultured hiPSC-derived neurons into the mouse brain, where cells can fully integrate into the neuronal circuits and initiate action potentials, is known to rescue the cellular stress. Grafting human neurons in the mouse brain additionally enables the study of human neurons up to later stages and in a more physiological setting. Modelling ID using the existing in vivo xenotransplantation model might decode disease etiology to a further extent than in vitro models can [32,33].

Although none of these models can completely recapitulate the complexity of a human brain that is affected by ID, they do provide promising tools to study components of disease etiology and design therapies. The timeline of the described models is strikingly similar to human fetal brain development, recapitulating key milestones of fetal and early postnatal development [26]. Early stages of ID pathology that were previously difficult (if not impossible) to obtain can now be studied more accurately and into great detail. The combination of patient-derived models and gene editing to obtain isogenic controls allows the examination of specific genetic components in the underlying cause for ID. Furthermore, the models can be utilized in the screening of pharmaceutical compounds and permit the design of personalized approaches in the development of therapies. As these models are continuously advancing, they can beyond doubt contribute to the understanding of ID pathophysiology and the search for therapeutic interventions in the next years to come.

## Figures and Tables

**Figure 1 ijms-23-06476-f001:**
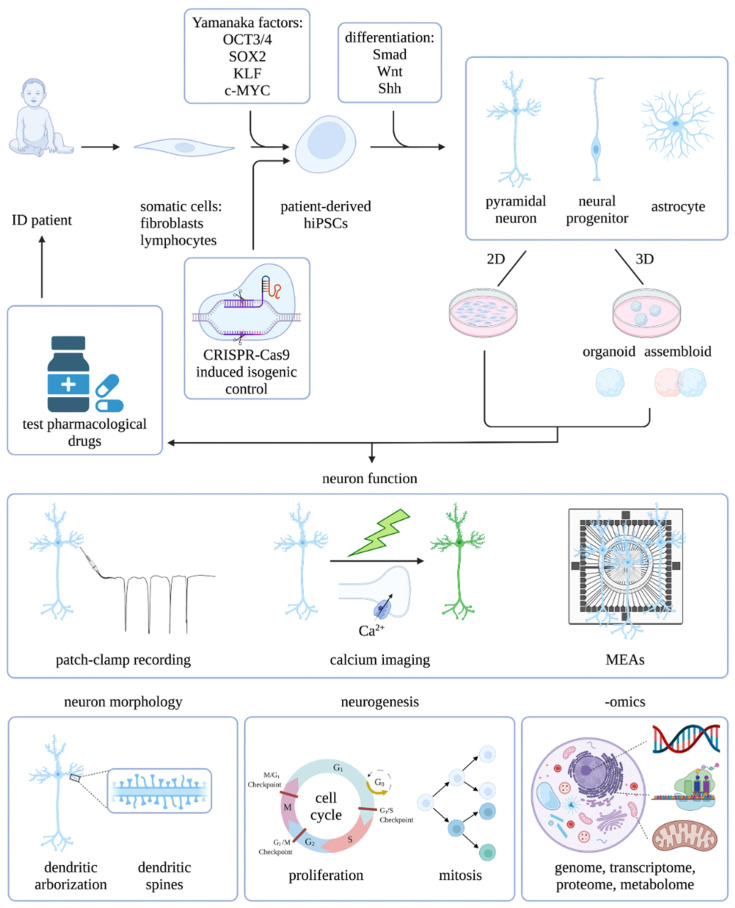
In vitro human brain models of ID. Patient-derived somatic cells are reprogrammed into iPSCs using Yamanaka factors (OCT34, SOX2, KLF, c-MYC). CRISPR-Cas9-induced isogenic controls can be obtained by reversing ID mutation. The hiPSCs can be differentiated into 2D neuronal cultures and into 3D organoids or assembloids. In vitro models can be applied to study neuron function (patch-clamp recording, calcium imaging or microelectrode arrays), neuron morphology, neurogenesis (e.g., proliferation and mitosis), and in multi-omics approaches. Testing pharmacological drugs on in vitro models can accelerate ID drug development.

**Figure 2 ijms-23-06476-f002:**
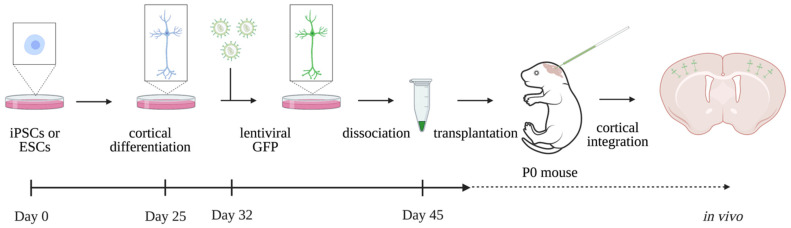
Experimental outline of in vivo xenotransplantation of human neurons. Human ESCs or iPSCs are differentiated in vitro into cortical neurons until day 45. Human neurons are transplanted into neonatal mice brains where neurons integrate into the cerebral cortex in vivo. Adapted with permission from Ref. [32]. © 2019, D. Linaro et al.

## Data Availability

Not applicable.

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
