# Peer review of "Human Brain Models of Intellectual Disability: Experimental Advances and Novelties"

_ijms, 2022, doi:10.3390/ijms23126476_

Round 1

Reviewer 1 Report

The authors describe multiple scientific breakthroughs during the past decades that have enabled the development of novel ID models and describe the situation in a few well known genetic syndromes.

ref nr 2 is from 2006, I think there are more recent publications on the diagnostic yield in patients with ID.

In 4. "Challenges and future perspectives" the authors mention still large problems and that it will cost a lot and will require a lot of time and expertise to improve the ID models. Here I miss discussion on the usefulness of different aspects of these developments and if the authors find it realistic that therapies will be designed for ID and in what period. 

Author Response

Thank you for reviewing our manuscript. We appreciate the suggestions that you provided and took these into account in our updated manuscript.

We now replaced reference 2 of the first version of the manuscript by two more recent references (see below) to give a more precise estimate of the current diagnostic yield of neurodevelopmental disorders, among which ID.

  1. Maia N, Nabais Sá MJ, Melo-Pires M, de Brouwer APM, Jorge P. Intellectual disability genomics: current state, pitfalls and future challenges. BMC Genomics. 2021;22(1):1–17.
  2. Srivastava S, Love-Nichols JA, Dies KA, Ledbetter DH, Martin CL, Chung WK, et al. Meta-analysis and multidisciplinary consensus statement: exome sequencing is a first-tier clinical diagnostic test for individuals with neurodevelopmental disorders. Genet Med. 2019;21(11):2413–21.

In section 4. “Challenges and future perspectives” we included a paragraph to describe the usefulness of the models explained in this review and what the future beholds of therapies that can be designed for ID.

In addition, we updated the manuscript on a few aspects using the suggestions of other reviewers of our manuscript. We now included the purpose of the review in the abstract and updated the reference list as the reference manager had made minor errors in the first version of the manuscript.

Reviewer 2 Report

This manuscript is written well, however, this need to revise just a little bit.

1) Please be more specific about the purpose of this review.

2) Could you give us more specific instructions on how to choose this citation?

3) Could you confirm list of references again? For example, is the citation in the following literature correct?

Bhaduri A, Andrews MG, Mancia Leon W, Jung D, Shin D, Allen D, et al. Cell stress in cortical organoids impairs molecular 469
subtype specification. Nature. 2020;

Author Response

Thank you for reviewing our manuscript. We appreciate the suggestions that you provided and took these into account in our updated manuscript.

1. The purpose of our manuscript is to provide an overview of the existing literature on scientific breakthroughs (Section 2) over the past decades that have been advancing the way ID can be studied in the human brain.

    • This is important because these breakthroughs have contributed to the now existing human brain models for ID, which have the potential to accelerate the identification of the underlying pathophysiological mechanisms and the development of therapies.
    • Before these models existed, studying ID mainly relied on animal models or post-mortem human brain samples. Although these models have proven to be useful, there are also a number of pitfalls in these models:
      • Animal models do not completely recapitulate the human pathophysiology which complicates the translatability of preclinical findings to humans.
      • Post mortem samples of ID are scarce in number and only provide the endpoint of the disease. Early developmental timepoints can therefore not be studied accurately in the human brain. Early timepoints are crucial to correctly understand ID pathophysiology as during early stages neurons are born and brain circuits become established. Therefore, the disease onset can already have started during early stages and it is important to understand what defects origin from these early timepoints which can be partially recapitulated the discussed models of the manuscript.
    • In addition to describing the different types of human brain models that have been established, we also wanted to describe examples of how these models are already applied for specific types of ID (Section 3).

We now mentioned the purpose of this review in the last sentence of the abstract of the paper. We hope this fulfills the first suggestion of your revision.

2. Could you help us by explaining which citation you mean here? Is it the citation on the left side of the first page of the paper? It is not completely clear to us to which citation you are referencing to. In case it is the citation on the first page on the left, we now included all the information that is known to us in the updated manuscript.

3. We double checked the reference list and indeed in the reference that you mentioned by Bhaduri A et al. there was information missing which we unfortunately missed in the first manuscript. The reference has now been updated in our reference manager and the reference list of the manuscript. Additionally, we double-checked the other references that we cited.

In addition, we updated the manuscript on a few aspects using the suggestions of the other reviewers of our manuscript. We now replaced reference 2 of the first version of the manuscript by two more recent references to give a more recent estimate of the current diagnostic yield of neurodevelopmental disorders, among which ID. In section 4 “Challenges and future perspectives” we included a paragraph to describe the usefulness of the models explained in this review and what the future beholds of therapies that can be designed for ID.

Reviewer 3 Report

This review article discussed the advent of induced pluripotent stem cells (iPSCs) and gene editing tools such as CRISPR/Cas9 used in modeling intellectual disability (ID). Currently, review articles must present the transparent reporting of the reviewing procedures. The manuscript used an old style of review. I would suggest the authors refer to the guideline for current literature reviews, such as PRISMA (http://www.prisma-statement.org/) and Cochrane Library (https://www.cochranelibrary.com/ ).

Author Response

Thank you for reviewing our manuscript. We appreciate the suggestions that you provided and took these into account during the revisions of our manuscript. We read about the PRISMA statement and Cochrane library, which are two sources of excellent guidelines for systematic reviews. The “instructions for authors” section on the website of the International Journal of Molecular Sciences (IJMS), to which we submitted our manuscript, describes how the PRISMA guidelines should be followed for systematic reviews. However, because our manuscript is a literature review instead of a systematic review, we followed the general instructions for reviews, which mentions that “Reviews should provide a complete and balanced overview of the latest progress in a given area of research”. We have done the latter for our manuscript and therefore hope that you agree that the suggested guidelines are not required for the type of review we have written in the manuscript.

Please notice that we updated the manuscript on a few aspects using the suggestions of the other reviewers of our manuscript. We now included the purpose of the manuscript in the abstract. We updated the reference list as the reference manager had made minor errors in the first version of the manuscript. We replaced reference 2 of the first version of the manuscript by two more recent references to give a more precise estimate of the current diagnostic yield of neurodevelopmental disorders, among which ID. In section 4 “Challenges and future perspectives” we included a paragraph to describe the usefulness of the models explained in this review and what the future beholds of therapies that can be designed for ID.

Round 2

Reviewer 3 Report

Review articles must present the transparent reporting of the reviewing procedures.

I have not read this manuscript show how they select these article to review and how they organize these articles.

This manuscript is a resubmission of an earlier submission. The following is a list of the peer review reports and author responses from that submission.